# DNA-Binding Protein Dps Protects *Escherichia coli* Cells against Multiple Stresses during Desiccation

**DOI:** 10.3390/biology12060853

**Published:** 2023-06-14

**Authors:** Nataliya Loiko, Ksenia Tereshkina, Vladislav Kovalenko, Andrey Moiseenko, Eduard Tereshkin, Olga S. Sokolova, Yurii Krupyanskii

**Affiliations:** 1Winogradsky Institute of Microbiology, Fundamentals of Biotechnology Federal Research Center, Russian Academy of Sciences, 117312 Moscow, Russia; 2Semenov Federal Research Center for Chemical Physics, Russian Academy of Sciences, 119991 Moscow, Russia; quebra-mola@yandex.ru (K.T.); vladislavkovalenko785@gmail.com (V.K.); postmoiseenko@gmail.com (A.M.); ramm@mail.ru (E.T.); yuriifkru@gmail.com (Y.K.); 3Faculty of Biology, Lomonosov Moscow State University, 119234 Moscow, Russia; sokolova184@gmail.com

**Keywords:** DNA-binding protein Dps, multiple desiccation stresses, *Escherichia coli*, DNA–Dps crystals, molecular dynamics methods, bacterial survival

## Abstract

**Simple Summary:**

The ability of bacteria to adapt to various types of stress has been studied for decades. It is directly related to the problem of nosocomial infections and the success of biotechnological processes involving bacteria. This study shows that bacteria can survive after being in a desiccated state for three months. The Dps protein, formed in cells to protect DNA, plays a major role in their survival. Bacteria with more of this protein in their cells have an advantage. Applying molecular dynamics methods made it possible to explain the protective function of Dps. The obtained results of the research will help create technologies for the conservation of useful bacteria used by humans.

**Abstract:**

Gradual dehydration is one of the frequent lethal yet poorly understood stresses that bacterial cells constantly face in the environment when their micro ecotopes dry out, as well as in industrial processes. Bacteria successfully survive extreme desiccation through complex rearrangements at the structural, physiological, and molecular levels, in which proteins are involved. The DNA-binding protein Dps has previously been shown to protect bacterial cells from many adverse effects. In our work, using engineered genetic models of *E. coli* to produce bacterial cells with overproduction of Dps protein, the protective function of Dps protein under multiple desiccation stresses was demonstrated for the first time. It was shown that the titer of viable cells after rehydration in the experimental variants with Dps protein overexpression was 1.5–8.5 times higher. Scanning electron microscopy was used to show a change in cell morphology upon rehydration. It was also proved that immobilization in the extracellular matrix, which is greater when the Dps protein is overexpressed, helps the cells survive. Transmission electron microscopy revealed disruption of the crystal structure of DNA–Dps crystals in *E. coli* cells that underwent desiccation stress and subsequent watering. Coarse-grained molecular dynamics simulations showed the protective function of Dps in DNA–Dps co-crystals during desiccation. The data obtained are important for improving biotechnological processes in which bacterial cells undergo desiccation.

## 1. Introduction

The lethal stress that bacterial cells face in the environment when their habitats dry up is desiccation [1]. Thus, soil bacteria constantly experience diurnal and seasonal variations in relative humidity, and during drought, they undergo complete dehydration [2]. In addition, desiccation is one of the main causes of cell death during biotechnological production processes or the storage of microbial cultures in collections [3,4].

During slow natural desiccation, bacteria experience several types of stresses at once: starvation, osmotic, pH, and dehydration stresses, in which membranes, proteins, and nucleic acids are damaged [5]. For many microorganisms, such exposures are fatal. However, some successfully survive extreme desiccation due to a complex set of interactions at the structural, physiological, and molecular levels [6], which involve RpoS, LexA, RecA, ArcA, Csps proteins, etc. [7,8,9,10].

DNA-binding protein Dps is involved in the protection of bacterial cells from various types of stresses, primarily oxidative and starvation stress, as a result of which the protein copy synthesis increases to 150–200 thousand [11,12,13,14,15,16,17,18,19,20,21,22]. It has been shown that it protects bacteria from heat and alkaline shocks, high pressure, toxic effects of heavy metals, UV and gamma radiation, and other types of negative effects [13,15,18,23,24,25,26,27,28].

The main task of the Dps protein is to participate in the process of DNA crystallization, called “in cellulo crystallization, bio- or nanocrystallization. [17,23,29,30,31,32,33]. Dps also prevents fenton-mediated oxidative damage by trapping hydroxyl radicals within the protein envelope and can oxidize transition metal ions by accumulating their oxides within its cavity [34,35]. Moreover, this ferritin-like protein (belongs to a new branch of the superfamily of ferritin-like proteins called mini ferritins) can rapidly oxidize divalent iron ions and subsequently accumulate clusters of trivalent iron ions in its nano cells, providing bacteria with physical and chemical protection [36]. To counteract various stresses, the cell requires both Dps activities: DNA-binding and ferroxidase, which are biochemically separable but act together to maintain DNA integrity and cell viability [15,22].

Dps protein was first detected in *E. coli*-starved cells in 1992 [37]. Later it was shown that *E. coli* Dps is a very compact and stable multifunctional protein (about 80–90 Å in diameter) consisting of 12 identical subunits with a flexible and lysine-rich N-end protruding from the dodecamer [38]. When the subunits are assembled, a spherical hollow cavity (about 40–50 Å in diameter) is formed, which serves as a compartment for iron storage [34,38,39]. Dps-similar mini ferritins are found in most known bacteria and archaea and reveal a striking degree of structure conservativity, which is related to their function: a hollow cuboid 90–100 Å wide with rounded corners and P23 symmetry [39,40]. Characterization of these proteins has been described for more than 60 microorganisms, and several thousand homologous genes have been annotated in modern genomic databases [41].

Despite numerous studies on the stress-protective properties of Dps protein, there is little data on its involvement in cell protection against osmotic stress and desiccation. Thus, Weber et al. found an increased expression of Dps during osmotic stress but could not determine its role during desiccation [42]. Additionally, scientists from the University of Wisconsin-Madison, using mutant strains of *Escherichia coli* O157: H7, showed that Dps is involved in cell protection during desiccation and osmotic stress, but only in the case of bacterial growth on Luria–Bertani broth with 12% NaCl (LB-12) at 37 °C [43].

Nevertheless, understanding the mechanisms of microbial resistance to desiccation and the role of proteins such as Dps is an important problem of cell biology. It is also important and promising for solving the problems of bioengineering of desiccation-sensitive cells [5]. 

Therefore, this study aimed to investigate the stress-protective role of the Dps protein in the experience of multiple desiccation stress by *E. coli* cells. The objectives of this work were: (1) to subject a population of *E. coli* bacteria with different content of Dps protein to long-term desiccation stress in the natural growth environment and to compare their adaptation responses to this exposure; (2) based on experimental data and molecular modeling methods, to identify the main mechanisms of cell protection during extreme desiccation, in which DNA-binding protein Dps takes part. To achieve these objectives, two engineered strains of *E. coli* were used to produce cells with overproduction of the Dps protein.

## 2. Materials and Methods

### 2.1. Bacteria

The objects of this study were the Gram-negative bacteria *Escherichia coli* Top10/pBAD-Dps (hereinafter Top10) and *E. coli* BL21-Gold(DE3)/pET-Dps (hereinafter Gold) from the Biotechnology Research Center collection [17,44,45]. These strains are genetic constructs that contain plasmids containing a DNA region encoding the Dps protein, allowing the production of cells with overproduction of Dps protein.

### 2.2. Cultivation

Bacteria were grown in Luria–Bertani (LB) (Broth, Miller, VWR, Radnor, PA, USA) medium with 150 µg/mL ampicillin. The inoculum, a steady-state growth phase culture (overnight culture), was added in an amount of 1 mL per 50 mL of medium (2%), which provided an initial optical density (OD) of 0.3 (λ = 540 nm, l = 10 mm; Spectrophotometer 7315, Jenway, Stone, Staffordshire, UK). Cultivation was carried out in 250 mL glass flasks with cotton plugs with 50 mL of the nutrient medium under stirring (120 rpm) and at 28 °C for 1 day.

To obtain bacteria in which the amount of Dps protein was significantly higher than normal, protein expression was induced by adding 1 g/L arabinose in the case of *E. coli* Top10 or 1 mM lactose in the case of *E. coli* BL21-Gold to cultures of strains of the linear growth phase (4 h). At the same time, the amount of Dps protein in cells of the Top10 strain increased 4-fold or more by the stationary phase of growth, and in cells of the BL21-Gold strain by 8-fold or more [17].

### 2.3. Obtaining Dehydrated Cell Preparations

Bacteria were grown on LB medium as described previously for 1 day. Each variant was obtained in five replicates using identical flasks and cotton plugs. The flasks closed only with cotton plugs through which natural evaporation occurs, were then stored without stirring at 23 °C to allow the process of natural drying of the cells in their own growth medium. The amount of culture fluid (CF) in the flasks was measured periodically, and the viability of the bacteria was examined by direct counting with Live/Dead dye and by the colony-forming unit (CFU) method when seeding on agar medium. Storage was performed for 6 months.

### 2.4. Watering of Cell Preparations

50 mL of distilled sterile water was added to the flasks with dried biomass. Shaking the flasks achieved complete resuspension of the flask contents. After a certain amount of water was added, aliquots were taken to determine the bacterial abundance and to prepare samples for electron and scanning microscopy.

### 2.5. Cell Count Determination

The total number of cells in the microbial fractions was determined by direct counting in an Axioplan fluorescent microscope (Carle Zeiss, Oberhausen, Germany) using fluorescent dye Live/Dead, (Biotium, Fremont, CA, USA), which allows counting “live” and “dead” cells (with impaired barrier function of the cytoplasmic membrane).

Cell viability was determined by the number of colony-forming units (CFU) when cell suspensions were seeded from appropriate dilutions on LB agar medium (with the addition of 1.5% Bacteriological Agar, Helicon, Moscow, Russia).

### 2.6. Cell Reactivation

An aliquot (1 mL) of the sample was placed in 10 mL of distilled sterile water and incubated under constant agitation for 1 day.

Microscopic observations to monitor the purity of the bacterial culture and the physiological state of the cells were performed using a Reichert microscope (Austria) with a phase-contrast device.

### 2.7. Scanning Electron Microscopy (SEM)

To obtain images of the rehydrated cells, the precipitate was collected after centrifugation of the samples (7000× *g*, 10 min). The precipitate was dehydrated in ethyl alcohol solutions of increasing concentration (15 to 100%) and transferred to 100% acetone. To image completely dehydrated cells, a sample of dehydrated contents was collected from the bottom of the flask and immediately transferred to 100% acetone. All samples were then washed twice with 100% acetone and dried at the critical point using a special chamber Hitachi Critical Point Dryer HCP-2 (Hitachi, Tokyo, Japan). The dried preparations were applied to a substrate with carbon tape, and then a thin layer of metal (gold/palladium) was sprayed on to create a conductive coating. The obtained samples were examined with the Scanning electron microscopeTM3000 (Hitachi, Tokyo, Japan) (accelerating voltage 15 kV) and the scanning electron microscope JEOL, JSM-6380LA (Jeol, Tokyo, Japan) (accelerating voltage 20 kV. SEI mode).

### 2.8. Transmission Electron Microscopy (TEM)

Cells were fixed with 2% glutaraldehyde for 5 hrs and postfixed with 0.5% paraformaldehyde; washed with a 0.1 M cacodylate buffer (pH = 7.4), contrasted with a 1% OsO4 solution in a cacodylate buffer (pH = 7.4), dehydrated in an increasing series of ethanol solutions, followed by dehydration with acetone, impregnated, and embedded in Epon-812 (following manufacturer’s instructions). Ultrathin sections (100–200 nm thick) were cut with a diamond knife (Diatome) on an ultramicrotome Ultracut-UCT (Leica Microsystems), transferred to copper 200 mesh grids, covered with formvar (SPI, Washington, DC, USA), and contrasted with lead citrate, according to the Reynolds established procedure [46]. For the analytical electron microscopy study, contrasting was omitted in some cases.

Ultra-thin sections were examined in transmission electron microscopes JEM-1011 and JEM-2100 (Jeol, Tokyo, Japan) with accelerating voltages of 80 kV and 200 kV, respectively, and magnification of ×13,000–21,000. Images were recorded with Ultrascan 1000XP and Orius SC1000D CCD cameras (Gatan, Pleasanton, CA, USA).

### 2.9. Co-Crystal Molecular Models

3D structure of the *E. coli* Dps protein nanocrystal (15 Dps molecules) was cut out from a crystal PDB ID: 6GCM using UCSF Chimera [47]. The N-terminal amino acid residues of each Dps subunit were added manually [48].

The DNA sequence corresponds to the part of this gene, and it has an increased affinity for the Dps protein [49]: 5′-GCACTATATTATGGGGTGATGGATATTCATGTCACGCCCCAAAATTAACTGAGTTCACCTAAACAGAAAGGATATAAACATCAGACAGGTTTACGTTACTATCAGGCATATCACCTCAGAATCAGATGAAAACTATAAAGAAATATCTATTATGGTTTTAATATTTG-3′. One double-stranded DNA molecule was inserted into each of the three mutually orthogonal channels [48,50] to obtain co-crystal. As a control, the dynamics of the same DNA molecule in solution were studied.

Co-crystals were prepared for simulations in GROMACS [51] using the coarse-grained force field MARTINI 2.1_DNA [52]. The DNA–Dps co-crystals and DNA molecules were placed in a cubic periodic box 1.3 × 10^5^ nm^3^. As the number of water molecules decreased from 10^6^ to fully dehydrated (solvated ions only), the volume of the cell gradually decreased to 3.2 × 10^4^ nm^3^. Three types of charged particles were used as ions, namely 15,516 Q+ (Na+ model), 100 Q++ (Ca++ and Mg++ model), and 14,000 Q−(Cl− model). These particles correspond to solvated ions in Martini v2.0 ions.itp topology file. The initial concentration of ions was chosen to correspond to the physiological one [53]. The ion concentrations of the simulated systems are given in Table 1. To avoid the artificial freezing of water associated with the peculiarities of the force field [54], 10% of water molecules in the systems were replaced by “water-antifreeze” particles.

### 2.10. Molecular Dynamics (MD)

Minimization and relaxation of the systems were carried out according to the protocols we developed earlier [55]. After that, MD simulations were carried out in periodic boxes at constant pressure and temperature (NPT ensemble). The integration step was 10 fs, and the trajectory length was 0.1 µs. The temperature of 300 K was maintained using a Langevin thermostat (the time constant was 1 ps). The pressure was maintained using a Parrinello–Rahman barostat (time constant was 4 ps, pressure was 1 bar, isothermal compressibility of water was 3.0 × 10^4^ bar^−1^). The cutoff radii for the Coulomb and van der Waals interactions were taken to be 1.2 nm. The dielectric constant of the medium was equal to 15 to provide implicit screening. Calculation time was chosen in such a way as to ensure that the graphs of the investigated spatial and energy characteristics of the systems reach a plateau.

### 2.11. Statistical Methods

All experiments were performed in triplicate. Statistical analysis was carried out using standard mathematical methods (Student’s t-test and calculation of the standard deviation) using the Microsoft Excel program. The data group was considered homogeneous if the mean square deviation σ did not exceed 10 percent. The differences between the data groups were considered valid under the probability criterion *p* < 0.05.

## 3. Results

### 3.1. Effect of Dps on the Viability of Desiccated of E. coli Top10 and E. coli BL21-Gold Cells

#### 3.1.1. Changing Medium Parameters during the Desiccation Experiment

Control (without Dps induction) and experimental (with Dps induction) bacterial cells of the stationary growth phase of two strains of *E. coli* Top10 and BL21-Gold were subjected to a process of slow natural desiccation in the growth medium at 23 °C. Moisture evaporation occurred naturally, which simulated natural processes. Figure 1 shows the changes in parameters during the desiccation. During long-term storage, the volume of the culture medium (CM) decreased (Figure 1a), the NaCl content in the medium increased (Figure 1b, calculated values), and the titer of viable cells decreased (Figure 1c,d). Thus, after 30 days of storage, the volume of the CM decreased by 12 mL (i.e., 34%), while the NaCl concentration increased to 0.67%. At this time, the bacterial cells were starved and began to experience osmotic stress [56] (Figure 1b). Complete desiccation of CM occurred 82 ± 3 days after the start of the experiment (Figure 1a). Ten days before this, the concentration of NaCl in CM reached 4%, and five days before this, the formation of salt crystals was observed.

#### 3.1.2. Decrease in Viable Cell Titer during the Desiccation Experiment

At the start of the desiccation experiment, the number of cells (detected by the CFU method) in the control and experimental populations of both strains was about 10^9^ cell/mL (Table 2). Moreover, in the experimental variants, the cell titer was lower because overproduction of the Dps protein leads to lysis of part of the population cells, which was shown earlier [43]. In the process of storage, autolysis started in cultures of both variants, which initially proceeded more intensively in experimental populations (with Dps activation), which also seems to be related to the imbalance of cell metabolism under conditions of protein oversynthesis (Figure 1c,d). However, after some time: for strain Top 10–30 days, for strain BL21-Gold—35 days, the situation changed, and the number of viable cells determined by CFU in experimental variants began to exceed the number of cells in control variants.

At a certain point in the experiment, when the volume of CM decreased by more than half, cell viability by the CFU method ceased to be detected, which was apparently due to the transition of cells to a non-culturable state [57]. In the Top 10 strain, this occurred after 45 days in the control variant and after 50 days in the experimental variant, while in the BL21-Gold strain, it occurred after 55 and 60 days (Figure 1c,d). Parallel direct cell counting with Live/Dead dye staining to distinguish between dead (stained orange) and live (green) cells confirmed this assumption. The titer of live cells determined by this method 60 days after the beginning of the experiment in all variants was higher than 10^7^ cells per ml (Table 2).

After the complete evaporation of CM, flasks with dehydrated cell samples were stored under the same conditions for a further 98 days (for a total experiment time of 6 months).

#### 3.1.3. Viability of Rehydrated Cells

Six months after the beginning of the experiment, flasks with experimental and control cultures were watered by replenishing the original volume of CM (50 mL of distilled H_2_O was added). In 2 h after watering, the number of viable cells was determined by two methods: direct count using Live/Dead dye (Figure 2) and the CFU method by the number of bacteria-producing colonies on a dense medium (Table 3). It turned out that the number of bacteria surviving after desiccation and subsequent watering was rather high (10^5^–10^7^ cells/mL), which comprised from 0.6% to 19.2% of the number of viable cells in the population 60 days after the beginning of the experiment (Table 2, bottom line). At the same time, in the experimental variants with Dps protein overexpression, the titer of surviving bacteria was higher than in the control ones: 11-fold for the Top10 strain; 3-fold for the BL21-Gold strain (Table 3). However, not all cells formed colonies on a dense medium (Table 3). About 0.006% of cells germinated in the experimental population of the Top10 strain, whereas in the control population without Dps protein overexpression, no growth was observed on a dense medium. In the case of the BL21-Gold strain, the colonies in the control and experimental populations yielded approximately the same number of cells: 0.02%. It should be noted that the colonies on a dense medium were very small in size (microcolonies), characteristic of the germination behavior of stressed bacterial cells [58]. To increase the number of cells yielding colonies on an agarized medium, reactivation procedures were performed. For this purpose, cell aliquots (1 mL) were transferred into 10 mL of sterile distilled water and incubated with constant agitation for 24 h. This contributed to the release of accumulated autoregulators of anabiosis from the cells [58]. As a result, it was possible to increase the number of cells yielding colonies on agarized medium by more than an order of magnitude and identify them in the control population of strain Top10.

Thus, the experiment showed that the Dps protein protects bacteria during desiccation stress, and its increased cell content allows them to survive.

### 3.2. Changes in Cell Morphology and Ultrastructure during Desiccation and Subsequent Watering

#### 3.2.1. Changes in the Morphology of Desiccated Cells Revealed by SEM

Scanning electron microscopy was used to assess the morphological changes in the cells during desiccation and subsequent watering (Figure 3). Studying images of completely dried cells showed that such exposure resulted in a change in cell morphology. They decreased in size by a factor of two or more and lost their regular bacilliform shape. The single dried cells detected had a flattened round or drop-shaped shape (Figure 3a). However, more common were bacterial conglomerates resembling a clump of shapeless cells and dense extracellular matrix (Figure 3b,c). In the Dps-activated experimental sample, larger matrix formations of a reticulate texture were more frequent than in the control sample, with cavities containing well-preserved bacterial cells of (0.3–0.5) × (0.5–0.7) µm in size (Figure 3d). This proves that the extracellular matrix guarantees cell survival under extreme conditions of total dehydration. Normally, the matrix comprises bacterial waste products and residues of the lysed part of the population [59]. Since, according to the visual assessment made by SEM, the amount of matrix is greater in the experimental samples, it probably also contains Dps protein, which is several times greater in induced cells. When part of the population dies as it is dried out, the protein escaping from the lysed cells becomes part of the extracellular matrix. The matrix is a kind of immobilization material for the bacteria, capable of keeping the cells intact during dehydration.

After rehydration, the cells recovered their regular bacilliform shape, and their size increased to (0.5–0.7) × (0.7–1.4) µm (Figure 3e–h). However, more damaged cells were detected in the control samples (Figure 3f, marked by arrows) than in the experimental samples (Figure 3h). When watered, the matrix also changed in structure, becoming a spongy or reticulate substance (Figure 3g). Importantly, such changes occurred to the cells just a couple of hours after watering, indicating a flexible adaptation system in the bacteria.

#### 3.2.2. Changes in Cell Ultrastructure Revealed by Transmission Electron Microscopy

To understand the processes occurring in bacterial cells during desiccation and subsequent watering, changes in their ultrastructure were studied using TEM. As mentioned earlier, nucleoid-associated proteins are responsible for DNA conservation in bacterial cells [60,61], and the Dps protein has the main protective function during prolonged starvation [12,14,15]. We have previously shown that the organization of DNA- Dps in a long-term starved E. coli bacterial cell is represented in most cases by three structures: nanocrystalline, liquid-crystalline, and nucleosome-like [17]. Moreover, the first one is the most common. In starved E. coli Top10 cells with activation of Dps protein expression, nanocrystals of large size, occupying up to half of the cell volume, were mainly detected. In E. coli BL21-Gold starved cells, nanocrystals were small, and their number varied from 2 to 7 [17]. All of them were highly ordered structures formed by correctly arranged Dps globules with DNA strands inside [17,50]. As part of the problem described in this study, it was important to determine what happens to such DNA-protecting nanocrystalline structures in the cell after desiccation stress.

In the first months of our experiment, until the amount of water in the samples became critical, the bacterial cells in the control and experimental samples experienced starvation stress. TEM investigation of such cells incubated for 30 days in both control and experimental samples revealed changes in their external morphology: a decrease in size, compression of protoplasts, and curvature of the cell shape (Figure 4a). All three types of previously described DNA–Dps organization structures were also detected in the cells, but most frequently, nanocrystals like those described earlier (Figure 4b). TEM examination of bacteria rehydrated after complete desiccation showed that already 30 min after watering, they practically recovered their morphological shape in both control and experimental samples (Figure 4c,d). These results agreed with SEM data showing that cells dramatically decreased in size and lost their regular rod-like shape when desiccated, but already in 2 h, it was completely restored. However, many lysed cells were detected in the populations, and their number in control samples without Dps protein overexpression was significantly higher, which agreed with the data on determining the titer of viable bacteria (Figure 4g, Table 3). The ultrastructure of the rehydrated bacteria was varied, but there were almost no cells with a clearly formed nanocrystal like those that starved cells formed before desiccation. In *E. coli* BL21-Gold samples (especially in the experimental populations), a small number of cells with very small (50–100 nm) crystals could be found (Figure 4e,f). Interestingly, the same individual crystal structures were also found in the intercellular space in completely lysed cells (Figure 4g). This suggests their stability and indirectly confirms our earlier conclusion that DNA–Dps crystals can independently participate in genetic information transfer even when the bacterial cells have already died [45].

The presence of periodicity in the detected crystal regions was proved by a calculated Fourier transform in which clear peaks related to the periodic structure were observed (shown in the lower right corner of Figure 4b,e–g). Unequal numbers of peaks at Fourier transform images on 4e-g probably occur from differences in the crystal’s size, orientation, and number of crystallographic defects. In this paper, we did not aim to study in detail crystal structure and periodicity, as it was the topic of a separate study.

In the experimental samples of *E. coli* Top10 rehydrated after desiccation, most of the intact cells contained dark stained regions similar to the previously described nanocrystals (Figure 4h). In about half of the cases, such regions were combined with the liquid-crystal DNA stacking located in the middle of the cell, as described previously (Figure 4c) [17]. However, Fourier transform of such regions showed that there were no periodic formations (Figure 4h). We tried to show by indirect signs that such regions most likely correspond to an intracellular crystal that has lost its periodic structure under desiccation stress. To obtain clearer images with TEM, OsO4 contrasting agent is used for sample preparation. It enters the region of intracellular crystal to a lesser extent due to the higher density of this structure; thus, such areas are lighter in the negatives and darker in the positives (see Figure 4b). We analyzed the detected areas with presumed “damaged” crystal structure and showed a decrease in signal intensity in them, the same as in the case of the crystal (Figure 5). Apparently, such regions with increased density relate to broken intracellular crystals as a consequence of dehydration.

Thus, desiccation stress leads to “breakage” of the DNA–Dps crystal structure, which does not lose its functionality as the majority of the population survives.

### 3.3. Explaining the Mechanisms of the Stress-Protective Action of Dps Protein during Cell Desiccation by Methods of Classical Molecular Dynamics (MD)

To explain the mechanisms of the stress-protective action of the Dps protein during cell desiccation, the method of classical molecular dynamics was used. Since it was shown using TEM that DNA–Dps crystals formed in cells exposed to stress factors during slow drying lose their structure after watering, it was interesting to understand in detail the processes taking place using MD methods. It was necessary to find out what conformational changes occur with DNA and DNA–Dps nanocrystals during desiccation (decrease in the water content in the system), accompanied by an increase in the concentration of ions.

We performed simulations for five types of systems with different amounts of water (both DNA in solution and DNA–Dps co-crystals): non-dehydrated system, 100% H_2_O (W_100_); the first stage of desiccation—75% H_2_O (W_75_); the second stage of desiccation—50% H_2_O (W_50_); the third stage of desiccation—25% H_2_O (W_25_); the fourth stage of desiccation—10% H_2_O (W_10_); the fifth stage—there is only bound H_2_O (W_0_), modeled by a hydration shell of the ions (Figure 6). The systems were calculated with a constant number of ions; however, when going from W_100_ to W_0_, their concentration increased significantly due to a decrease in the number of water molecules.

#### 3.3.1. Systems W_100_ and W_75_

The simulations showed that, at a sufficient degree of water cut in the W_100_ and W_75_ systems, the DNA–Dps nanocrystals retain their structure and are almost identical. The N-terminal regions of Dps molecules are responsible for the binding of DNA molecules to Dps, which was already shown earlier in our work [55]. In this case, the regulation of DNA binding to Dps is carried out by divalent ions and depends on the ionic composition of the medium [62]. A slight increase in the NaCl concentration with a decrease in the number of water molecules by 25% has practically no effect on the potential energy of DNA binding to Dps. Only small changes in energy within ±5% are observed. DNA inside the crystal adapts to the internal environment and adjusts to the shape of the channels, which is consistent with previous results [63]. The terminal regions of DNA lying outside the channels, as well as free DNAs in solution, do not undergo significant conformational changes (Figure 6).

#### 3.3.2. System W_50_

With a decrease in the amount of water in the system by 50%, desiccation effects begin to appear. The molecular diameter of Dps molecules and, accordingly, the distance between the molecules located in the two closest sites of the crystal lattice remain unchanged (Figure 7, curve 1). However, the internal structure of the nanocrystal begins to change. Approach and interaction of Dps molecules, which lie side by side but did not contact each other, take place; the crystal becomes denser (Figure 7, curve 2). At the same time, crystal defects appear. These defects were observed repeatedly during the simulations. Then they either increased with dehydration, leading to the detachment of individual Dps molecules or a part of the crystal (spontaneous fragmentation of the crystal) or were eliminated. Kinks and unwindings appear in DNA molecules in solution and their free termini outside the co-crystal. At the level of individual base pairs, there is a transition of DNA from the B-form to the A-form. However, inside the crystal, the DNA is stabilized and adapted to the internal environment of the channels, maintaining a linear shape.

#### 3.3.3. Systems W_25_ and More Desiccated

The core of the crystal, formed by internal Dps molecules, continues to condense as the amount of water in the system decreases below 25%. At this stage, condensation occurs both due to the approach of Dps molecules to one another and a change in the outer diameter (compression) of the dodecamers. It should be noted that the dimensions of the periodic box remain sufficient to adopt the entire crystal and the surrounding water so that the compression of molecules occurs due to internal causes, and not through the action of a barostat. Additionally, at this moment, detachment of some surface Dps molecules occurs, which has a stochastic character. The compression of Dps molecules is caused by the redistribution of water and ions in the computational cell. The initial content of water molecules inside the dodecamer turns out to be slightly reduced and corresponds to a general decrease in the water content in the system. However, unlike systems with higher water content, at W_25,_ the redistribution of water and ions is so critical that the potential energy of interaction Dps-ions (E_i_) becomes much closer in value to the energy Dps-water (E_w_). If at W_100_ and W_75,_ the following relation E_i_ ≈ 0.15E_w_ is valid, and at W_50_ E_i_ ≈ 0.23E_w_, then at W_25_ and W_10,_ the ratio is much higher and is approximately equal to E_i_ ≈ 0.5E_w_. The predominant binding of ions to the Dps surface leads to water expulsion from the surface and the compression of molecules. DNA became

The compression of dodecamers can already be observed at the transition from W_100_–W_75_ to W_50_. However, the most critical changes occur with further desiccation (W_25_ and below). Protein structural compactness can be observed by such a parameter as the gyration radius, which depends on the squared distances from the center of mass to the i-th atom and the masses of the atoms. The gyration radius of Dps molecules during desiccation decreases from ~3.8 nm to 3.5–3.55 nm (Figure 8a). In this case, one can see some difference in the degree of compaction of protein molecules located in the center of the nanocrystal and on its periphery. At W_50_, compression affects, to a greater extent, the molecules at the periphery of the crystal. Further desiccation (W_25_) leads to a decrease in the gyration radius for all protein molecules but affects the central molecules to a greater extent. Thus, at different degrees of desiccation, one can observe inhomogeneity in the behavior of the crystal.

A further decrease in water does not affect Dps molecules but causes significant conformational changes in free DNA regions. The degree of DNA compaction depending on the water content in DNA–Dps systems is shown in Figure 8b. At W_25_, DNA strands begin to bend and shrink (DNA in solution) or adsorb on the crystal surface (DNA termini in a co-crystal). A high salt concentration promotes the transition of DNA from the B-form to the A-form. However, in addition to A-B transitions, DNA bends, wrong linking, and unwinding are observed. Due to the faster binding of the DNA termini to the crystal, DNA loops begin to form on the surface of the crystal. The part of DNA lying inside the DNA–Dps crystal, on the contrary, is strongly stabilized, which ensures the safety of its structure. At W_10_ and below, the number of kinks in free DNA regions increases. In addition to kinks, there is a divergence of DNA chains. Maintaining the native DNA structure becomes possible only with dense packing in Dps channels.

## 4. Discussion

The problem of bacterial survival, to which this article is devoted, is one of the most topical issues in modern microbiology. On the one hand, we need to learn how to kill microorganisms so that they do not spoil food, metal structures, industrial equipment, etc., and do not cause the development of pathogenic processes in human and mammalian organisms. On the other hand, we need to learn how to maintain the viability of “useful” microorganisms that help to produce food and medicines, participate in soil enrichment and remediation, maintain the health of the gastrointestinal tract, etc. Any scientific research in this direction adds something to the overall body of knowledge, which will then be used for practical purposes.

The first conclusion of this study was another confirmation that bacteria have such flexible adaptation systems that they can survive in the most unlikely conditions. In the experiment presented, they had to face a long period of starvation, an increase in NaCl salt in the CM (up to a saturated solution), extreme desiccation, and storage in this form for more than three months. Additionally, after these tests, a significant part of the bacterial population (up to 20%) survived anyway. Bacteria can go uncultivated, shrink in size, change their morphology, and still survive. That is, they survive just fine on dry surfaces for long periods when we think it is impossible.

Methods to detect bacterial contamination on surfaces or in liquid media often show negative results [64,65,66]. However, this result is erroneous. The bacteria may be unculturable, and additional resuscitation procedures are needed to detect them. In our study, it was shown that when using the CFU method to count viable cells, the result was severely underestimated. Another method of direct cell counting with Live/Dead dye showed a much larger number of viable cells. Reactivation procedures also helped identify viable cells [58]. For example, simple washing of the cells in distilled water, which promotes the release of anabiosis factors, allowed the detection of a greater number of viable cells. It is likely that by using other reactivation methods, we could improve this indicator even more. This is why it is so difficult to defeat nosocomial infections, which cause great problems in hospital treatment [67]. It is especially important to consider the adaptation abilities and features of the bacteria shown in this study when developing new techniques for their detection [68]. Under natural conditions, when microorganisms can immobilize natural carriers, their ability to survive stressful conditions, including desiccation, is multiplied [69].

The second conclusion from the work was that it was possible to show that the survival of bacteria under multiple desiccation stress depends on the amount of Dps protein in the cells. Artificially increasing its content by overexpression from the plasmid allowed more bacteria to survive this severe stress. This was performed both by additional protection of DNA inside the cell and by increasing the amount of extracellular matrix on which immobilization allowed the cells to survive. However, desiccation is not without consequences for the bacteria. The DNA–Dps crystals formed in the cells during starvation lose their ordered structure but continue to fulfill their DNA-protecting function. This is the third important conclusion drawn from the work. Additionally, interesting is the fact that individual DNA–Dps crystals were found to persist in the dead cells. They proved resistant to the action of cell lytic enzymes and survived the effects of desiccation. This suggests they are extremely resistant and can participate in horizontal gene transfer when no intact cells remain.

The use of molecular dynamics methods made it possible to confirm that in the process of gradual dehydration, the internal structure of the nanocrystal begins to change and compact. At first, there is a convergence and interaction of nearby, but not contacting each other DPS molecules, and crystal defects arise, leading to the detachment of individual DPS molecules. Subsequently, water molecules, which were inside the cavity of the dodecamer, come out, and the dodecamer is compressed, causing a decrease in the characteristic radius of gyration of DPS molecules from ~3.8 nm to 3.5–3.55 nm. The DNA molecule undergoes more critical changes.

The DNA molecule undergoes more critical changes during dehydration. It is known that DNA is a remarkably adaptable molecule that can undergo major conformational rearrangements without irreversible damage [70]. It can support many different shapes depending on the base sequence and environment [71,72]. A solution of high ionic strength (increased NaCl concentration during desiccation) promotes the transition of DNA from the B-form to the A-form. However, in addition to A-B transitions, DNA bends, mismatches and knockout of nucleotide residues are observed for DNA outside the crystal. In addition, DNA breaks can occur. Whereas the DNA termini in the co-crystal are adsorbed onto its surface, and the part of the DNA that remains inside the crystal is well stabilized and adapted to the channel environment, which ensures that its structure remains intact. On the contrary, DNA strands left in the drying solution without DPS support bend and shrink, losing functionality. This important point explains the mechanism of DNA conservation via Dps protein during stress.

The results obtained can be applied to developing new technologies for the conservation of beneficial bacteria used by humans in biotechnological processes, especially those subjected to desiccation. By selecting strains with higher productivity (or creating special genetic constructs) on the Dps protein, the problem of bacterial mortality in production cycles with harsh conditions can be solved. Additionally, our study clearly showed that bacteria survive even after being in a dehydrated state for 3 months. This should be considered by sanitary and medical workers, and more stringent methods of disinfecting instruments and hospital surfaces should be used.

## 5. Conclusions

The results obtained in this work are an important addition to our knowledge of the ability of bacteria to adapt to complex stress situations. Studies have shown that *E. coli* bacteria can survive after a slow desiccation for 6 months. Experiencing starvation stress, osmotic stress, and dehydration stress simultaneously, they can retain up to 20% of the half-life. DNA-binding protein Dps plays a large role in bacterial survival, preserving DNA during extreme dehydration. Cells with higher amounts of this protein have a greater chance of survival. DNA–Dps crystals formed in the cell during starvation, after the onset of complete desiccation and subsequent dehydration, basically lose their crystal structure but continue to perform their function. DNA protected by the DPS protein retains its structure, while DNA in solution bends and shrinks, losing its functionality. The results obtained are important for understanding the adaptive mechanisms that allow bacteria to overcome the most difficult stressful effects, and for creating new methods for detecting and eliminating bacterial contamination on surfaces in medical institutions and the food industry. Using strains with higher productivity of the Dps protein in biotechnological processes will help solve the problem of bacterial death in production cycles.

## Figures and Tables

**Figure 1 biology-12-00853-f001:**
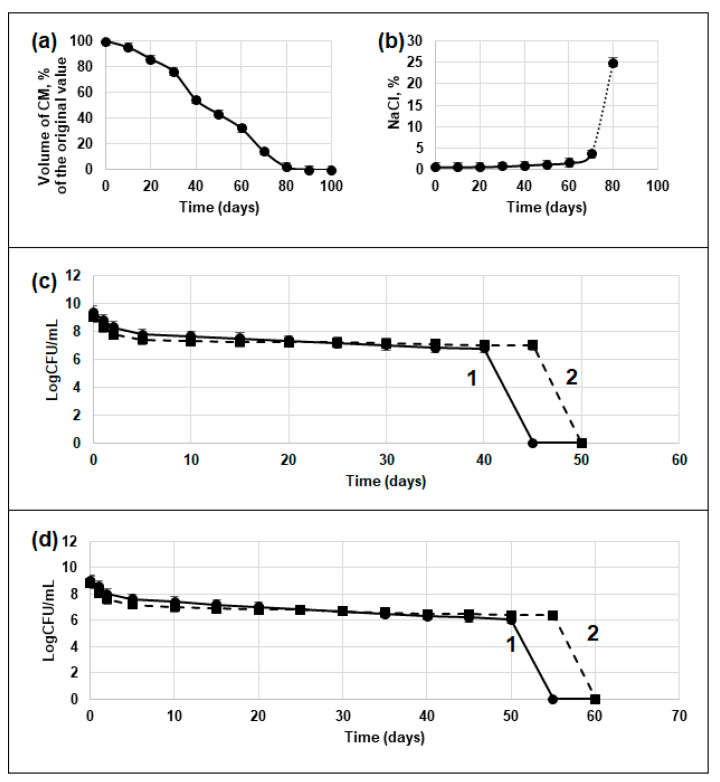
Characteristics of the desiccation (are the same for both strains Top10 and BL21-Gold): (**a**) decrease volume of culture medium (CM); (**b**) increase in NaCl concentration; decrease in cell viability (CFU) during storage: (**c**) strain Top10; (**d**) strain BL21-Gold. 1—control population without activation of Dps synthesis; 2—experimental population with activation of Dps synthesis.

**Figure 2 biology-12-00853-f002:**
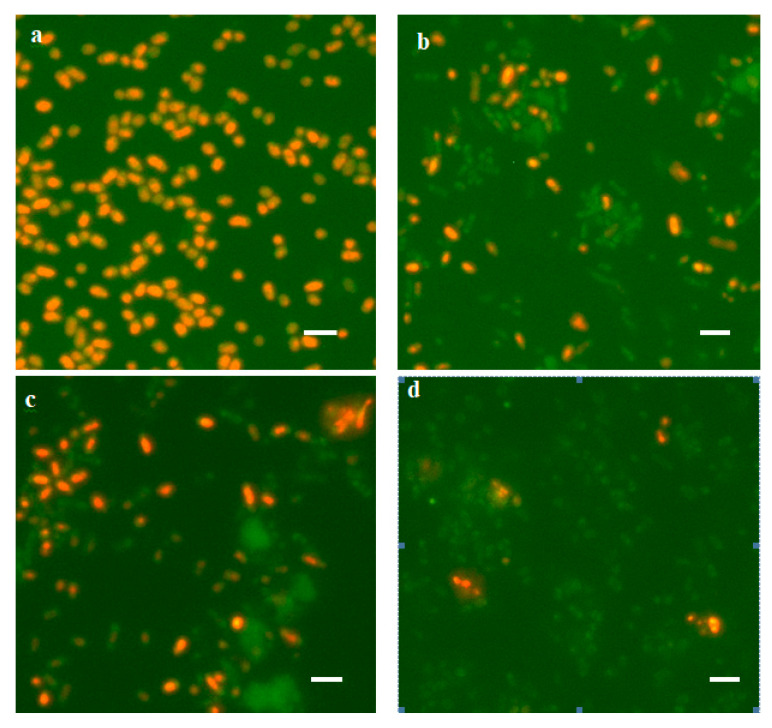
View of rehydrated (2 h) cells stained with Live/Dead dye: (**a**)—Top10 without Dps activation; (**b**)—Top10 with Dps activation; (**c**)—BL21-Gold without Dps protein activation; (**d**)—BL21-Gold with Dps activation. Dead cells are colored orange, live cells are colored green. Scale marker 2 µm.

**Figure 3 biology-12-00853-f003:**
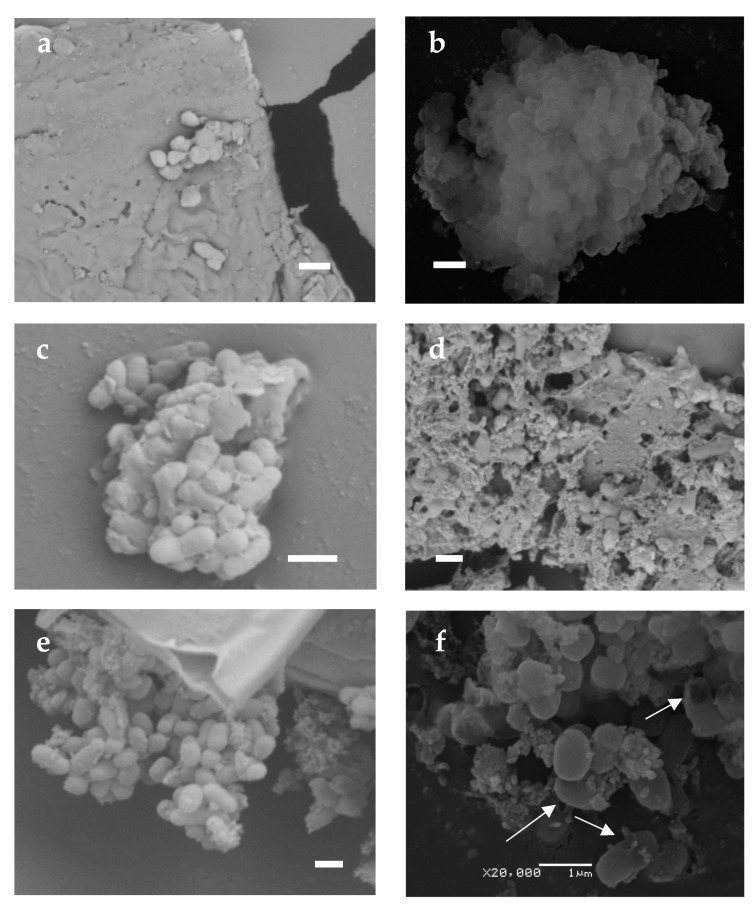
SEM view of BL21-Gold strain cells: (**a**,**b**,**e**,**f**)—control populations without Dps protein activation; (**c**,**d**,**g**,**h**)—experimental populations with Dps protein activation; (**a**–**d**)—desiccation cells; (**e**–**h**)—cells after 2 h watering. The arrows in (**f**) show cell damage. (**a**,**b**); (**c**,**d**); (**e**,**f**) and (**g**,**h**) refer to the same samples, but differ in image scale. Images (**a**,**c**–**e**,**g**) were obtained with the TM3000 scanning electron microscope; images (**b**,**f**,**h**) were obtained with the JEOL scanning electron microscope, JSM-6380LA. Scale marker 1 µm.

**Figure 4 biology-12-00853-f004:**
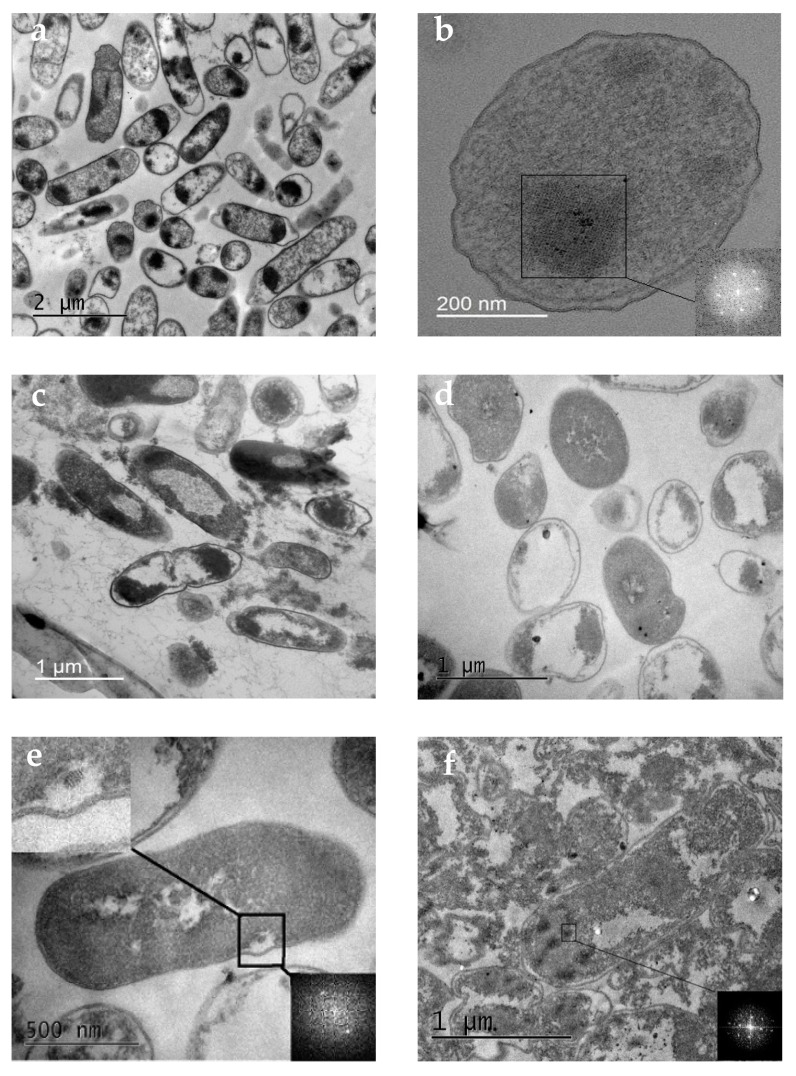
TEM view of E. coli Top10 (**a**–**d**,**h**) and E. coli BL21-Gold (**e**–**g**) cells: (**a**,**d**)—starved cells for 30 days; (**c**–**h**)—rehydrated (30 min) cells after desiccation. (**a**–**c**,**f**,**h**)—bacteria with Dps protein overexpression; (**d**,**e**,**g**)—bacteria without Dps overexpression. The lower right-hand corners of (**b**,**e**–**h**) show the Fourier transforms.

**Figure 5 biology-12-00853-f005:**
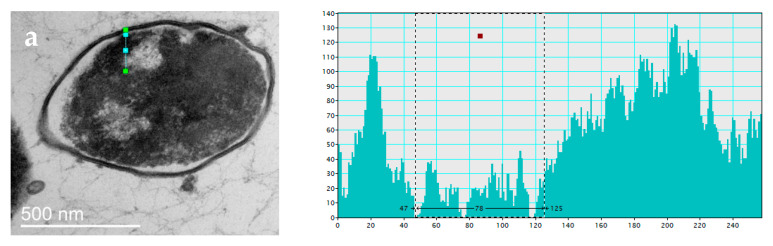
TEM view of *E. coli* Top10 (**a**,**c**) and *E. coli* BL21-Gold (**b**) cells: (**a**,**b**)—rehydrated (30 min) cells after desiccation; (**c**)—starved cells for 30 days. Profiles of signal intensity located on the right part of the figure show an intensity decrease, which we associated with the densest part of the cell, DNA–Dps crystal, then it still is not destroyed by desiccation. (**a**,**b**) Intensity profiles of the densest area inside the cell exposed to desiccation stress, which we assumed once were DNA–Dps crystal; (**c**) intensity profile of the DNA–Dps crystal. Intensity profiles are related to the region depicted on the left figure as a cyan line, cyan dots in the middle of the line related to dashed frame on the right figure.

**Figure 6 biology-12-00853-f006:**
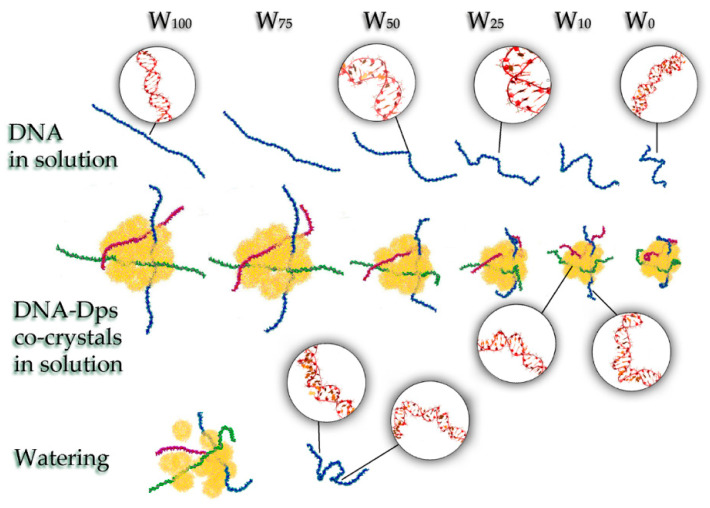
Changes in DNA conformation and DNA–Dps crystal structure in systems with different water contents.

**Figure 7 biology-12-00853-f007:**
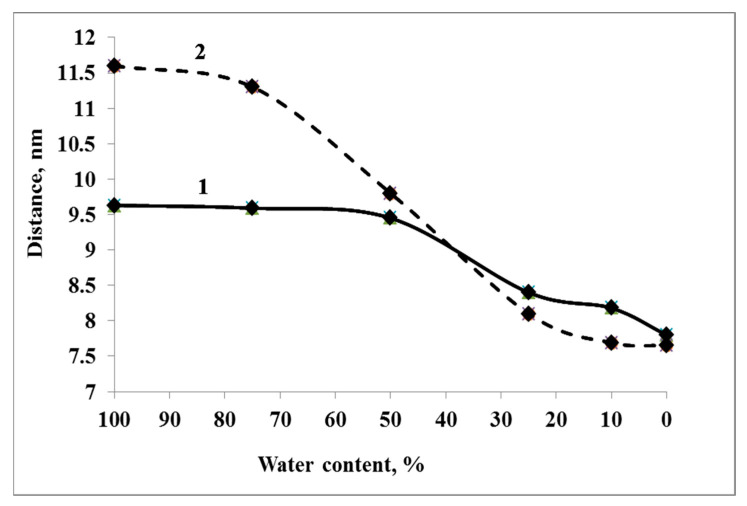
Change in the distance between the centers of mass of neighboring Dps dodecamers, lying in the nearest nodes of the crystal lattice (1). The distance between the centers of mass of Dps molecules that are not directly in contact with each other in the native system (W_75_ and above), but which acquire contact during desiccation (W_50_ and below) due to structural rearrangements of the crystal (2).

**Figure 8 biology-12-00853-f008:**
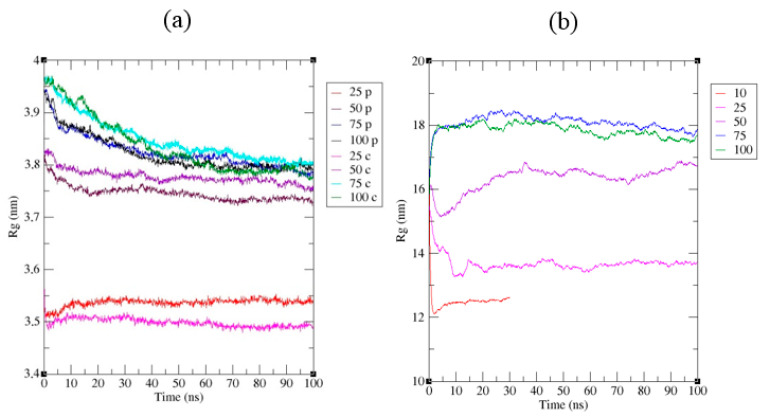
Changes in the gyration radii of DPS (**a**) and DNA (**b**) molecules as a function of time at different water contents in the system. The numbers in the legends indicate the percentage of water, indices: c—DPS molecule in the center of the crystal, p—DPS molecule at the periphery of the crystal.

**Table 1 biology-12-00853-t001:** Ion concentrations at different stages of dehydration, mM. The name of the systems indicates the percentage of water.

Name/Ion	Na^+^	Ca^++^	Cl^−^
W100	199.2	1.3	179.7
W75	254.5	1.6	229.6
W50	368.0	2.4	332.1
W25	658.7	4.2	594.3
W10	822.1	5.3	741.8
W0	1252.4	8.1	1130.0

**Table 2 biology-12-00853-t002:** Changes in cell titer in control and experimental bacterial populations during the desiccation process.

Time of Experiments	Method	Titre of Cells, Cell/mL			
		*E. coli* Top10		*E. coli* BL21-Gold	
		*Without Dps* *Activation*	*With Dps* *Activation*	*Without Dps* *Activation*	*With Dps* *Activation*
Start of experiment	by CFU	(2.5 ± 0.5) × 10^9^	(1.3 ± 0.1) × 10^9^	(1.0 ± 0.1) × 10^9^	(7.6 ± 0.6) × 10^8^
30 days	by CFU	(1.0 ± 0.1) × 10^7^	(1.5 ± 0.1) × 10^7^	(5.0 ± 0.3) × 10^6^	(5.1 ± 0.3) × 10^6^
	total score with L/D	(5.4 ± 0.2) × 10^8^	(6.9 ± 0.4) × 10^8^	(1.9 ± 0.1) × 10^8^	(2.6 ± 0.2) × 10^8^
60 days	by CFU	0	0	0	0
	total score with L/D	(6.1 ± 0.5) × 10^7^	(8.3 ± 0.6) × 10^7^	(3.3 ± 0.2) × 10^7^	(4.9 ± 0.3) × 10^7^

**Table 3 biology-12-00853-t003:** Cells titer of two strains after 2 h rehydrated.

Strain	Dps Activation	Titre of Cells, Cells/mL			
		By the CFU Method		Total Score with Live/Dead	
		*W* *ithout Reactivation*	*After* *Reactivation*	*Number of All Cells*	*Number of Living Cells (in %)*
Top10	-	0	(3.6 ± 0.2) × 10^2^	(2.1 ± 0.2) × 10^6^	(3.8 ± 0.2) × 10^5^ (18.1)
	+	(2.4 ± 0.2) × 10^2^	(8.2 ± 0.5) × 10^3^	(4.9 ± 0.3) × 10^6^	(4.2 ± 0.2) × 10^6^ (85.7)
BL21-Gold	-	(6.4 ± 0.3) × 10^2^	(3.5 ± 0.2) × 10^3^	(4.5 ± 0.3) × 10^6^	(3.2 ± 0.2) × 10^6^ (71.1)
	+	(2.2 ± 0.1) × 10^3^	(9.7 ± 0.7) × 10^3^	(9.7 ± 0.6) × 10^6^	(9.4 ± 0.2) × 10^6^ (96.9)

## Data Availability

Not applicable.

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
