# Peer review of "DNA-Binding Protein Dps Protects Escherichia coli Cells against Multiple Stresses during Desiccation"

_biology, 2023, doi:10.3390/biology12060853_

Round 1
Reviewer 1 Report
The paper " Dna-Binding Protein Dps Protects Escherichia coli Cells Against Multiple Stresses During Desiccation" is well written and the subject is interesting. The introduction and Discussion can be improved. Results were presented clearly. The methodology is appropriate.
May be accepted for publication after minor revisions.
Introduction
The statement regarding the aim of the work can be improved and specified. I suggest detailing the aim of the work and the main steps undertaken to reach this goal to evidence the differences respect with other studies working on desiccation and Dps protein. In this regard, at the end of the Introduction section it is reported: “Despite numerous studies on the stress-protective properties of Dps protein, there is little data on its involvement in cell protection against osmotic stress and desiccation”. However, a few studies have been published regarding Dps role in desiccation, for example, LeBlanc et al. 2008.; Stasic et al., 2012; Hu et al., 2017. Since this topic is extensive, I would recommend changing this sentence and highlighting the main aims of this interesting work.
I also suggest including the explanation of using two different recombinant strains and the main features and reasons for including both in the study.
Results:
Line 224: 109 kl/ml is not clear the meaning of the units.
Figure 2: I suggest pointing out also in the caption the difference in dye color between live and dead cells.
In general, reduce the reference to literature data to comment results within Result section and add them in the Discussion. An example at lines 270-1: "This contributed to the release of accumulated autoregulators of anabiosis from the cells [56]."
Discussion:
As a general comment: I would suggest underlining the main results that distinguish your work from others in addition to the well-reported conclusions since they are closer to summary than discussion in some points.
Lines 550-557: on the basis of last comments on possible applications, do you think that the obtained results can wide the potential applications towards diverse Gram-negative bacteria?
Minor comments:
- Lines 33-33: I suggest rephrasing to be clearer.
- Verify in all the text that E. coli is written with a space between the genus and the species.
- Correct in all text the way the temperature is reported: 370C is not correct the symbol for degree or it is underlined; it should be 37 °C (examples are present at lines 78, 99, 205).
- Line 96: not clear the meaning of used symbol for weave length.
- Line 116: "2.5 The cell count determination" revise into "2.5 Cell count determination "
- Line 257: correct the type of brackets for the reference.
- Line 369: it seems to have an extra comma that is not necessary.
- Lines 370-1: a verb seems missing after “there”. Correct accordingly.
- Line 560: correct to E. coli
Reviewer 2 Report
Introduction is lacking data about Dsp in regads to E. Coli metabolism
Line 110: how was the viability of the bacteria examined?
Line 127: "Microscopic observations" what exactly was under observation?
Line 135: "using a special chamber" provide details in order to make the work reproducible.
Overall it's a good paper, I therefore suggest only a minor revision
Author Response
Dear Reviewer. Thank you very much for your attention to our work. All your comments have been taken into account.
- Introduction is lacking data about Dsp in regads to E. Coli metabolism
Added text: «Dps protein was first detected in E. coli starved cells in 1992 [37]. Later it was shown that E. coli Dps is a very compact and stable multifunctional protein (about 80-90 Å in diameter) consisting of 12 identical subunits with a flexible and lysine-rich N-end protruding from the dodecamer [38]. When the subunits are assembled, a spherical hollow cavity (about 40-50 Å in diameter) is formed, which serves as a compartment for iron storage [34, 38, 39]»
- Line 110: how was the viability of the bacteria examined?
Added the phrase: «by direct counting with Live/Dead dye and by the colony-forming unit (CFU) method when seeding on agar medium».
- Line 127: "Microscopic observations" what exactly was under observation?
Added the assignment of microscopic examinations: «to monitor the purity of the bacterial culture and the physiological state of the cells»
- Line 135: "using a special chamber" provide details in order to make the work reproducible.
Added the name of the special camera: «Hitachi Critical Point Dryer HCP-2 (Hitachi, Japan)»
Reviewer 3 Report
The manuscript entitled “Dna-Binding Protein Dos Protects Escherichia coli Cells Against Multiple Stresses During Desiccation” deals with the overexpression of Dps and its ability and mechanism to combat multiple desiccation stresses. The study is exciting, and the authors have represented their findings well. However, the authors may focus on the following minor suggestions to improve the manuscript further.
· Fig 1a and 1b: Should present the Graph for both strains.
· Fig 3 legend: What is the difference between a & b, c & d, e & F, and g & h? If it is the difference in resolution, pl mention it clearly.
· The discussion section can improve by comparing the findings of some latest studies with this work. Also, need to emphasize the signification findings of this study.
· Should ass some possible applications of the finding of this study in the conclusion section.
Minor editing of English language required
Author Response
Dear Reviewer. Thank you very much for your attention to our work. All your comments have been taken into account.
- Fig 1a and 1b: Should present the Graph for both strains.
These graphs apply to both strains. An explanation has been inserted in the caption to the figure.
The curves shown in graphs 1a and 1b are identical for both strains. A corresponding clarification was made in the caption of Figure 1 to make it clear.
- Fig 3 legend: What is the difference between a & b, c & d, e & F, and g & h? If it is the difference in resolution, pl mention it clearly.
We added this note to the description of the figure.
3 The discussion section can improve by comparing the findings of some latest studies with this work. Also, need to emphasize the signification findings of this study.
Added to the discussion the importance of the results of the study
- Should ass some possible applications of the finding of this study in the conclusion section.
Added reasoning on the application of research results to the conclusion section
Comments on the Quality of English Language
Minor editing of English language required
Fixed bugs